# The Application of (+)-Catechin and Polydatin as Functional Additives for Biodegradable Polyesters

**DOI:** 10.3390/ijms21020414

**Published:** 2020-01-09

**Authors:** Malgorzata Latos-Brozio, Anna Masek

**Affiliations:** Institute of Polymer and Dye Technology, Lodz University of Technology, 12/16 Stefanowskiego Street, 90-924 Lodz, Poland

**Keywords:** polylactide (PLA), polyhydroxyalkanoate (PHA), natural polyphenols, (+)-catechin, polydatin, stabilizers, indicators, aging

## Abstract

Plant polyphenols are a huge group of compounds with a wide spectrum of applications. Substances from this group have been used in polymer materials such as stabilizers, dyes, indicators, fungicides, and bactericides, especially in new generation packaging materials. The aim of this study is to obtain environmentally friendly materials based on the biodegradable aliphatic polyesters, polylactide (PLA) and polyhydroxyalkanoate (PHA), with plant functional additives, (+)-catechin and polydatin. These natural polyphenols (polydatin and (+)-catechin) have not been used so far in polymer materials (especially in biodegradable polyesters) as stabilizers, dyes, and indicators of aging. The application of polydatin and (+)-catechin as multifunctional additives for biodegradable polymers is a scientific novelty. This paper presents the following analyses of polyester materials: SEM microscopy, wide angle x-ray diffraction, mechanical properties, thermal analysis, surface free energy analysis, and determination of change of color after controlled UV exposure, thermal oxidation and weathering. Both PLA and PHA polyesters were characterized by higher resistance to oxidation and greater resistance to degradation under the influence of UV radiation. In addition, (+)-catechin was used simultaneously as a dye and an indicator of the aging time of polymeric materials. In contrast, polydatin did not dye polymers, but was a very good indicator of their lifetime, changing color under the influence of various external factors. Both polyphenols can be successfully used as natural additives for pro-ecological polyesters.

## 1. Introduction

Thermoplastic aliphatic polyesters, polylactide (PLA) and polyhydroxyalkanoates (PHAs), are called “double green” polymers. These polymeric materials are fully biodegradable and produced from renewable raw materials [1,2,3]. PLA is obtained by condensation polymerization of lactic acid derived from corn and other plant root starches, sugarcane, and other resources [4]. Polymers from the polyhydroxyalkanoates group are biosynthesised by gram-positive and gram-negative bacteria as intracellular carbon and energy storage compounds [5]. Substances of plant origin are used more and more as functional additives for polymers, mainly biodegradable materials. The combination of biodegradable polymer matrix and modifying additives derived from natural raw materials seems to be very beneficial for both the environment and human health [6,7]. Such fully natural polymeric materials fit perfectly into the current trends of packaging materials and can potentially find wide applications in the packaging industry.

Compounds of plant origin deserve special attention because substances from this group have been proposed as stabilizers for polymeric materials, dyes, antibacterial and antifungal substances, freshness indicators of packaged food products, as well as indicators of the lifetime of polymers. Kirschweng et al. [8] presented the possibility of using carotenoids, vegetable oils, and polyphenols as natural stabilizers for polymers. Plant-derived dyes, such as colorants obtained from marigold petals (*Tagetes erecta L.*), rhubarb rhizomes (*Radix et Rhizoma Rhei*), turmeric rhizomes (*Curcuma longa Linn.*), garcinia bark (*Garcinia dulcis Kurz.*), sappan bark (*Caesalpinia sappan Linn.*), as well as catechu bark (*Acacia catechu Willd.*) were applied to dyeing PLA [9]. Plant-based dyes have also been successfully used as indicators in polymer packaging materials, especially in the new generation “smart packaging”. Curcumin, grape peel extract, beetroot extract, and anthocyanins have been proposed as indicators that change color as a result of pH changes that accompany food spoilage. These natural substances were used to monitor the freshness of cod and pork [10,11,12]. In addition, plant substances such as juglone and compounds from the flavonoid group (silymarin and quercetin) have been proposed as color indicators of aging for polymeric materials, biodegradable polyesters, as well as elastomers [13,14,15]. Plant substances, especially phenolic compounds, have strong bactericidal and fungicidal properties [16,17,18]. For this reason, they are increasingly used as agents against bacteria and fungi in active packaging for the food industry [19]. Furthermore, essential oils, such as rosemary oil (*R. officinalis L.*), caraway oil (*C. carvi L.*), and fennel oil (*F. vulgare Mill*) have been applied in PLA as antibacterial additives [20].

The aim of this study is to produce environmentally friendly materials based on biodegradable aliphatic polyesters, PLA and PHA, with the addition of (+)-catechin and polydatin as plant functional substances (stabilizers and dyes, as well as indicators of aging for polymeric materials). In the literature, there is no information about the application of (+)-catechin and polydatin as multifunctional modifying substances for biodegradable polymers. Polydatin, also called piceid (3,4’,5-trihydroxystilbene-3-β-D-glucoside), is found in grape skins, nuts, pomegranates, and other plants. The health-promoting properties of this polyphenol are known, such as: anticancer, anti-inflammatory, anti-allergy, and potential cardioprotective effects. Moreover, polydatin has strong antioxidant properties [21,22,23,24]. Catechins, like polydatin, are common in the plant world. They have been identified in fruit (apples, blueberries, kiwi, and strawberries), green and black tea, red wine, and cocoa. A rich source of catechins is green tea, in which these polyphenols occur in the form of (−)-epigallocatechin gallate, (−)-epicatechin, (−)-epigallocatechin, (−)-epicatechin gallate, and (+)-catechin. Catechins are known for their strong antioxidant properties and significant positive effects on human health [25,26,27,28,29,30]. Due to the very favorable and valuable properties (particularly antioxidant ability) of both polyphenols, they have been used as natural functional additives for pro-ecological biodegradable polyesters. The proposed polymeric materials can be a good alternative to commonly used petrochemical packaging. The advantage of the materials obtained is that they are made exclusively of natural substances, their recycling is easy, and they do not burden the environment with waste that is difficult to dispose of. The addition of plant substances as a stabilizer controls the lifetime of polymeric materials, and furthermore, the change in the color of samples with polyphenols can be an indicator of their aging time.

## 2. Results and Discussion

The study began with the evaluation of properties of the polymeric materials. The first step was the analysis based on evaluation of SEM images (Figure 1) of the structure of the materials. The SEM images showed topography of samples. Samples on PLA polymers were more compact and smoother than PHA samples. In contrast, samples based on PHA had a more porous structure than PLA. This more porous structure can affect the faster oxidation and aging of PHA. In addition, it was observed that the (+)-catechin was not visible in the SEM images of both polyesters. The second polyphenol, polydatin, was observed on the SEM image in the PLA matrix, but it was not present in SEM photograph of PHA/polydatin. 

The next stage of research was the analysis of mechanical properties of polyester materials with the addition of plant substances (Table 1). The addition of polydatin and (+)-catechin to biodegradable polymers did not change their mechanical properties. The properties of the reference PLA and PHA were similar to those of samples with polyphenols. The effect of controlled UV and thermo-oxidative aging, lasting 100 h, on the materials was also investigated. For PLA-based samples, a significant decrease (about 25%) in the tensile strength T_S_ was observed after aging. A decrease in this value suggests deterioration in the mechanical properties of the PLA samples. Contrasting results were obtained for PHA samples. Thermal oxidation and UV aging for 100 h caused an increase (about five times) in the tensile strength, T_S_, and a decrease (about three to five times) in the elongation at break, E_b_, of the samples. Such results could indicate an increase in the crystalline phase content of samples after aging. The appearance of a larger amount of crystalline phase in PHA materials could be the result of exposure of samples to elevated temperatures accompanying both UV aging at 60 °C and thermal oxidation at 70 °C). Further aging of the samples (200 h and 300 h) caused significant degradation, making it impossible to test the mechanical properties. 

On the basis of mechanical properties, the aging coefficient (K) of materials (Figure 2), after thermal oxidation and UV aging, was determined. The aging coefficient (K) was calculated from the following Equation (1):(1)$$K\;=\;\frac{\left(T_{S}\times E_{b}\right)after\;aging}{\left(T_{S}\times E_{b}\right)before\;aging}$$
where T_S_ is tensile strength (MPa) and E_b_ is elongation at break (%). 

The aging coefficient characterized the degree of material degradation. A value of K close to zero meant that the sample was more susceptible to aging. A value of K close to one meant that the sample was resistant to degradation.

The reference PLA and PHA polyester samples had lower K aging coefficients than samples with natural additives (K_(Termooxidation 100h) PLA_ = 0.52, K_(UV 100h) PLA_ = 0.51, K_(Termooxidation 100h) PHA_ = 0.87, and K_(UV 100h) PHA_ = 0.70). The PLA aging coefficients with polydatin and (+)-catechin were closer to one (K_(Termooxidation 100h) PLA/(+)-Catechin_ = 0.71, K_(UV 100h) PLA/(+)-Catechin_ = 0.76, K_(Termooxidation 100h) PLA/Polydatin_ = 0.93, and K_(UV 100h) PLA/Polydatin_ = 0.64) than the K coefficients of the reference samples. Similar results were obtained for the second tested polyester, PHA (K_(Termooxidation 100h) PHA/(+)-Catechin_ = 0.95, K_(UV 100h) PHA/(+)-Catechin_ = 1.12, K_(Termooxidation 100h) PHA/Polydatin_ = 1.05, and K_(UV 100h) PHA/Polydatin_ = 1.26). These results suggest that the addition of plant polyphenols increased the resistance of both polymers to thermo-oxidative and UV aging. 

The rheological properties were evaluated for one example of the polyesters, PLA. The addition of natural antioxidants to PLA did not significantly affect the rheological properties of the polymeric material on the basis of the melt flow index (Figure 3). The addition of plant polyphenols slightly reduced melt flow (MFI) and melt volume rate (MVR) index, i.e., about 12% for both types of sample. The decrease of MFI and MFR index of samples with polyphenols can be due to the H-bonding effect. Intermolecular hydrogen bonds (from polyphenols) bring the molecules closer together, as a result of which the density of the material increases. Lower values of melt flow index indicate an increase in the viscosity of the samples with natural substances. 

The next step was to determine the Vicat softening temperature of the samples (Figure 4). As with the MFI and MFR indices, the addition of natural substances did not significantly affect the Vicat softening temperatures of the polyesters. The impact of UV and thermal oxidation aging on this parameter, important in polymer processing, was also investigated. Controlled UV aging lasting 100 h did not affect changes in softening temperature. In contrast, thermal oxidation aging caused a significant increase in the Vicat temperature. It was found that the Vicat softening temperature for materials based on PLA increased by about 15% for reference PLA and about 30% for PLA with polydatin and (+)-catechin. The following increases in the Vicat softening temperature were found for PHA-based materials: about 117% for reference PHA, 65% for PHA/(+)-catechin, and 83% for PHA/polydatin. Higher Vicat softening temperatures suggest a higher crystalline phase content in the samples, which can increase as a result of the chemical crystallization process accompanying the aging processes of polyesters. Of the two types of polymers tested, a higher increase in the Vicat softening temperature was observed for PHA samples. This could be due to the higher crystalline phase content in the samples of this polyester, which was confirmed by the results of DSC and WAXD.

Table 2 and Figure 5 presents the results of differential scanning calorimetry tests. The addition of polydatin and (+)-catechin did not change the temperature ranges of the following sample phase changes: glass transition temperature (T_g_), crystallization temperature (T_cc_), and melting temperature (T_m_). The DSC curves of the PHA samples had two peaks corresponding to the melting of the crystalline phase, therefore, there are two values, T_m_ and ΔH_m_, in Table 2, for a single sample. The oxidation temperature is the temperature at which an exothermic oxidation peak appears on the DSC curve. Table 2 gives the initial (onset) temperatures of the oxidation peaks. A significant increase in the oxidation temperature (T_o_) was found for PLA by about 50 °C and for PHA by 7 to 20 °C. Higher oxidation temperatures of the PLA and PHA samples with polydatin and (+)-catechin would increase the stability of the materials and improve their resistance to oxidation. The results suggest that the polyphenols studied can be successfully used as natural stabilizers for polyesters.

In addition, the degree of crystallinity of the polymeric materials was calculated based on the DSC analysis. The crystallinity of polyester materials can be determined by DSC and knowledge of the melting enthalpy of 100% crystalline 100% polymer. The degree of crystallinity, X_c_, was calculated as follows: X_c_ (%) = ∆H_m_/∆H_0_ × 100, where ∆H_m_ was the melting enthalpy of the test polymer and ∆H_0_ was the melting enthalpy of 100% crystalline 100% PLA or PHA. According to literature data, the melting enthalpy estimated for 100% crystalline PLA is 97.3 J/g [31]. According to the manufacturer, the PHA polymer is P(3,4HB), therefore, the melting enthalpy of 100% crystalline PHB 146 J/g [32] was used as the melting enthalpy of 100% crystalline PHA. The value of the melting enthalpy of 100% crystalline PHB does not quite correspond to the value of melting enthalpy of 100% crystalline P(3,4HB) and the calculations contain a certain error resulting from these differences. However, the only value of melting enthalpy of 100% crystalline PHB 146 J/g is available in the literature as melting enthalpy for polymers from the PHAs group. The DSC curves of the PHA samples had two peaks corresponding to the melting of the crystalline phase, therefore, the melting enthalpy of PHA samples taken for calculations of crystallinity, was the sum of ΔH_m_ for the two peaks corresponding to the melting of the crystalline phase. The degree of crystallinity was 2.6% to 3.2% for X_cPLA_ and 23.5% to 27.7% for X_cPHA_. These results could be subject to error due to the assumed values of 100% crystallinity of the polymers. The calculated content of the crystalline phase was consistent with WAXD results. The addition of polydatin and (+)-catechin to PLA slightly increased the degree of crystallinity of the materials. Plant polyphenols can act as nucleating substances, increasing the crystallinity of PLA with an amorphous character. The addition of polydatin and (+)-catechin to crystalline PHAs prevented crystallization and reduced the degree of crystallinity of the materials.

Table 2 also shows the results of differential scanning calorimetry tests after thermal oxidation aging. As a result of aging, a significant decrease in enthalpy of oxidation (ΔH_o_) was observed. In addition, there was a shift in oxidation temperatures towards higher values. Thermal oxidation aging also caused changes in the crystallinity of polymers. The degree of crystallinity of all samples based on PHA increased by about 5% to 11% (PHA 5%, PHA/(+)-catechin 8%, and PHA/polydatin 11%), whereas for the PLA samples a clear increase in crystallinity was observed only for PLA/polydatin (about 85%). Samples of both polyesters with polydatin were characterized by a significant increase in the degree of crystallinity under the influence of high temperature (70 °C). The presence of polydatin in polyester samples could have been caused a greater increase in crystallinity than the presence of (+)-catechin. This could be due to the fact that polydatin has a higher molecular weight (M_w_ 390.38 g/mol) then (+)-catechin (M_w_ 290.27 g/mol) and can behave in the polymer matrix as nuclei.

The next step in the thermal analysis of samples was to determine whether the addition of natural polyphenols affects the thermal stability of the polyesters (Figure 6 and Table 3). Table 3 shows the values of T20, T50, and T90 for the PLA and PHA samples with polydatin and (+)-catechin, where T20, T50, and T90 refer to the loss of 20%, 50%, and 90%, respectively, of the initial mass of the sample as a function of temperature. The thermal decomposition of the PLA materials was two steps, but the addition of polydatin and (+)-catechin did not change the thermal stability, as evidenced by small differences in the T20, T50, and T90 values. In contrast to the PLA materials, thermal decomposition of PHA with plant polyphenols occurred in three steps. As with PLA samples, the addition of polyphenols to PHA did not significantly change the thermal stability. The T20, T50, and T90 values of PHA and PHA with polydatin and (+)-catechin differed by 1 to 6 °C.

Figure 7 shows the WAXD spectra of PLA (A) and PHA (B) with natural substances. The WAXD spectrum for PLA samples corresponds to the very low content of the crystalline phase (calculated on the basis of DSC results, X_cPLA_ = 2.6% to 3.2%) of this polyester and confirmed the amorphous nature of the PLA-based materials. On the WAXD diffractogram for PLA-based materials, only broad maxima of diffusive scattering occur, which indicates the existence of only close-range ordering at the molecular level of this polymer, and thus of the amorphous internal structure of the samples. The WAXD spectra for materials based on PHA polyester showed that the samples had a crystalline nature (X_cPHA_ = 23.5% to 27.7% calculated on the basis of the DSC results). The reference PHA was a crystalline polymer, as indicated by the sharp diffraction peaks found in the WAXD spectra. Three peaks were observed on the WAXD spectra of PHA, the most pronounced, with the highest intensity at 16.2°, and two smaller peaks at 12.9° and 18.3°. Moreover, the addition of plant substances, polydatin and (+)-catechin, reduced the crystallinity of PHA, as evidenced by the smaller number of sharp diffraction peaks found on the WAXD spectra. On the WAXD spectra of PHA/(+)-catechin only one peak was observed at 13.7°. Similarly, only one peak at 13.2° was found for the PHA/polydatin sample. In the case of both samples, peaks were characterized by a low intensity (Figure 7B).

The next step was to determine the surface free energy of the samples (Figure 8). The addition of polydatin and (+)-catechin caused a slight decrease in the surface free energy of the PHA samples and a significant decrease (about 30% PLA/(+)-catechin and 20% PLA/polydatin) in the surface free energy of the PLA-based samples. The changing nature of the surface of samples, especially materials based on PLA, could be the result of the interaction of numerous hydroxyl groups present in plant polyphenols with the hydrophilic surface of polymers.

The samples of polymeric materials were subjected to controlled UV aging. The surface free energy of the reference PLA increased after 300 h of UV irradiation, while the surface energy values of PLA samples with plant substances remained similar to those before aging. The increase of the surface free energy of reference PLA means sample degradation due to UV aging. No change in the surface energy value of PLA samples with (+)-catechin and polydatin testifies to the effective stabilizing effect of these natural polyphenols to stop the degradation processes of the polymer material. Similar results were obtained for samples based on PHA. It was found that the addition of polydatin and (+)-catechin protected the samples against UV aging for 100 h. After 300 h of UV aging, the surface free energy of PHA samples increased markedly as a result of material degradation. The differences between the stability of the two polyesters could be due to the greater resistance of the PLA polymer matrix to UV radiation. The higher resistance of PLA to UV aging could be due to the fact that this polyester is less susceptible to oxidation due to the radical mechanisms activated by UV radiation. The slower PLA oxidation could be related to the structure of material, which is more compact and less porous as compared with PHA (SEM photographs).

Figure 9 presents the potential of phytochemicals as dyes and indicator substances, showing the lifetime of polymeric materials. Plant-derived polyphenols are a huge group of compounds with diverse properties and are characterized by different colors, from colorless, through yellow and orange, to brown. This study shows the potential of two extreme-colored polyphenols, transparent-white polydatin and orange (+)-catechin. Polydatin did not change the color of the polymer materials, whereas the (+)-catechin gave the polyesters an orangish-yellow color. However, in the case of samples with the addition of both polyphenols, visible color changes were observed under the influence of various external factors. Figure 9 summarizes the change of color coefficients, dE·ab, with visual observations after 100, 200 and 300 h of UV exposure (A), thermal oxidation (B), and weathering (C). It is statistically assumed that, when the coefficient of color change, dE·ab, is 2 < dE·ab < 3.5, the difference of color can be seen by the average observer. The range of 3.5 < dE·ab < 5.0 means that there is a distinct color difference, while in the dE·ab > 5 range colors are perceived as completely different. Thus, the color changes after aging of the both polyesters, PLA and PHA, are noticeable by the average observer. The spectrophotometrically determined change of color coefficient, dE·ab, corresponds to the visual observations of samples after aging. All types of aging clearly changed the color of the samples, especially UV and weathering. For example, the PLA/polydatin sample under the influence of UV aging gradually changed color from colorless-transparent to yellowish-brown, and PHA/polydatin from milky to beige-brown. Samples of polyesters with (+)-catechin also had a very pronounced color change. The PLA/(+)-catechin sample changed from slightly yellow to an orangish-brown color, in proportion to the UV aging time and weathering also caused a very pronounced change in the color of the PHA/(+)-catechin sample, from slightly orange to a browish color. The color of samples with polyphenols changed due to the oxidation of these natural compounds under the influence of aging factors. Polyphenols absorb UV radiation, and therefore, they can provide protection of polymers against UVA and UVB radiation [22]. As a result of UV absorption and oxidation of polyphenols under the influence of degrading factors, there was a significant change in the color of these polyphenols, clearly visible in polymeric materials and indicating the lifetime of the materials.

On the basis of the polymeric materials presented in this study, a wide spectrum of possibilities for applying natural polyphenols has been shown. (+)-Catechin can be used both as a dye, giving polyesters a yellowish-orange color, as well as an indicator substance that changes color under the influence of external factors, such as UV radiation. In contrast, polydatin does not change the color of transparent PLA and milky PHA, but clearly changes the color of these materials when exposed to aging and can be used as a substance that indicates the lifetime of polyester materials.

A proposed mechanism of oxidation of PLA and PHA stabilized with polydatin and (+)-catechin is discussed below. The general mechanism of polymer degradation can be presented according to the following scheme (Scheme 1):

Polyester samples were subjected to various types of aging, i.e., UV, thermal oxidation, and weathering. Weathering aging is accompanied by the most factors causing polymer degradation which include the following: high temperature (thermal oxidation), UV radiation (UV aging), and humidity (hydrolytic aging).

Thermal degradation begins when the thermokinetic energy of interatomic vibration in the polymer chain is equal to or exceeds the energy of interatomic bond dissociation. Then, the interatomic bonds in the main polymer chain break, resulting in the formation of two active macroradicals at the ends of the shorter chains. These macroradicals take part in the next stages of polymer degradation.

When PLA and PHA polymers absorb the incident UV/VIS radiation, photodegradation can occur. Photodegradation is initiated when the energy of absorbed radiation is greater than or at least equal to the dissociation energy of individual bonds in the macromolecule. In addition, this type of degradation can be initiated when the polymers have chromophore groups in their structure that absorb UV/VIS radiation. In PLA and PHA, the chromophore groups are the C=O carbonyl groups. At room temperature, the polymer’s chromophore groups are in the basic singlet state. When the chromophore groups absorb the photons, they pass into the excited singlet stage, and in the next stage they can be transformed into the excited triplet state. The carbonyl groups in the induced triplet state are very reactive radicals. These reactive radicals are able to break the interatomic bond in the polymer backbone to form two macroradicals at the ends of the shorter chains. In addition, they can induce intramolecular transfer of a hydrogen atom to the formation of two shorter polymer chains. The generated macroradicals take part in further degradation processes of PLA and PHA materials.

In addition to elevated temperature and UV radiation, during weathering aging the samples are also affected by humidity that causes PLA and PHA hydrolytic degradation. During hydrolytic aging, water penetrates the polymer structure, causing hydrolysis of ester bonds (especially in the amorphous phase of PLA and PHA) and creating shorter polymer chains, i.e., oligo- and monomers. In the next stage of aging, water-soluble oligomers penetrate into the surrounding sample environment. On the one hand, if the oligomer release rate is faster than the water diffusion rate into the PLA and PHA polymer sample, surface erosion occurs. On the other hand, if the rate of water diffusion is greater than the release of oligomers, erosion occurs in the entire volume of the polymer material. When the products of hydrolytic degradation of PLA and PHA samples release very slowly from the depth of materials and at the same time increase the rate of hydrolysis (autocatalysis), then, the erosion of the sample core is accelerated.

Substances of plant origin, such as polydatin and (+)-catechin, are characterized by strong antioxidant properties, and therefore they can be used as natural antioxidants in polymers [8].

The task of natural substances in polymers is to eliminate chemical compounds that cause PLA and PHA degradation. Polydatin and (+)-catechin can be used to stabilize polymers as scavengers of free radicals, and also as antioxidants that prevent or delay the oxidation of polymers by reactions with radicals RO^●^, peroxy radicals (ROO^●^), as well as hydroperoxide groups (R-OOH).

## 3. Materials and Methods 

### 3.1. Reagents

The materials used in this study were two biodegradable polymers, polylactide (PLA) and polymer P(3,4HB) 2001 from the polyhydroxyalkanoate group of polymers (PHA). Polylactide (PLA), IngeoTM Biopolymer 4043D PLA, was produced by Nature WorksTM (Minnetonka, MN, USA) and had the following properties: T_g_ = 55 to 60 °C, T_m_ = 145 to 160 °C, and melt flow index MFI = 6 g/10 min. Polymer PHA was obtained from Simag Holdings LTD (Hong Kong, China) and had the following properties: P(3,4HB) containing 12 mol% 4-hydroxybutyrate, the average M_w_ was approximately 520 kDa, MVR = 15 to 20 g/10 min, (assay conditions of temperature 170 °C and nominal load 2.16 kg), and a density of 1.25 g/cm^3^.

Two plant-derived polyphenols were used as multifunctional polymer additives. Polydatin (≥95% (HPLC)) and (+)-catechin (≥98% (HPLC)) were obtained from Sigma-Aldrich (Steinheim, Germany). Structural formulae of polymers and polyphenols are shown in Figure 10.

### 3.2. Method for Preparation of PLA and PHA Samples with Natural Polyphenols

Dried (12 h, 50 °C) granulates of PLA and PHA were mixed with 1 part by weight of polydatin and (+)-catechin and extruded using a laboratory extruder and strip samples, with a thickness of 1.6 to 1.8 mm, were obtained. The temperature of the working chamber of the extruder was 180 °C for PLA and 160 °C for PHA, screw rotation speed = 40 rpm and extrusion pressure = 17 atm. 

### 3.3. Measurement Methods

#### 3.3.1. Scanning Electron Microscopy (SEM)

On the basis of the images obtained from the scanning electron microscope (SEM) LEO 1530 (Carl Zeiss AG, Oberchoken, Germany), the dispersion of polyphenols in a polyester matrix was evaluated. The test samples were prepared as follows: samples were fractured in liquid nitrogen and sputtered with carbon. Magnification was 1000 times.

#### 3.3.2. Wide Angle X-ray Diffraction (WAXD)

An X-ray diffractometer, Bruker model D2 Phaser (Bruker, Billerica, MA, United States), equipped with a Lynxeye detector, was used to study the crystal structure. The analysis was performed in the 2θ angle range of 2 to 40° with steps of 0.02°.

#### 3.3.3. Mechanical Properties

Determination of the mechanical properties was done using a Zwick Roell Z005 test machine (manufacturer Zwick Roell, Ulm, Germany). The measurement conditions were a preload of 0.1 N and a test speed of 50 mm/min. The following parameters were determined: T_Fmax_, the maximum stress [MPa]; E_Fmax,_ the elongation at break [%]; T_S_, the tensile strength [MPa]; and E_b_, the elongation at break [%]. For the mechanical tests, samples of extruded strips with a thickness of 1.6 to 1.8 mm and length of 150 mm were used. To determine mechanical properties, six test samples were cut out from each of three control samples.

#### 3.3.4. Surface Free Energy of PLA and PHA Samples

Tests were performed using an OEC 15EC goniometer (DataPhysics Instruments GmbH, Filderstadt, Germany). Surface free energy was determined by the method after Owens, Wendt, Rabel and Kaelble (OWRK). Polar and disperse contributions to the surface energy and surface tension, respectively, are combined by forming the sum of both parts. Therefore, one obtains Equations (2) and (3):(2)$$\sigma_{l}\;=\;\sigma_{l}^{d}\;+\;\sigma_{l}^{p}$$
(3)$$\sigma_{S}\;=\;\sigma_{S}^{d}\;+\;\sigma_{S}^{p}$$
where $\sigma_{l}^{d}$ and $\sigma_{l}^{p}$ represent the disperse and polar parts of the liquid, while $\sigma_{S}^{d}$ and $\sigma_{S}^{p}$ stand for the respective contributions of the solid. 

The interfacial energy can be calculated according to Owens, Wendt, Rabel and Kaelble from the contributions of the liquid and the solid by forming the geometric mean.

For $\sigma_{Sl}$ one obtains Equation (4):(4)$$\sigma_{Sl}\;=\;\sigma_{S}\;+\;\sigma_{l}\;-\;2\left(\sqrt{\sigma_{S}^{d}\cdot \sigma_{l}^{d}}+\;\sqrt{\sigma_{S}^{p}\cdot \sigma_{l}^{p}}\right)$$

Substituting this term for $\sigma_{Sl}$ in equation (1) and solving for the unknown quantities gives an equation of a straight line of the form Equation (5):(5)y = a · x + b
with Equation (6):(6)$$y\;=\;\frac{1+\cos\theta }{2}\cdot \frac{\sigma_{l}}{\sqrt{\sigma_{l}^{d}}};\;x\;=\;\sqrt{\frac{\sigma_{l}^{p}}{\sigma_{l}^{d}}};\;a\;=\;\sqrt{\sigma_{S}^{p}};\;b\;=\;\sqrt{\sigma_{S}^{d}}$$

Therefore, by plotting y versus x, $\sigma_{S}^{p}$ can be calculated from the slope of the fitted line and $\sigma_{S}^{d}$ from the intersection with the vertical axis. To achieve this, the contact angle of at least two liquids on the unknown solid must be determined. The measurements of the contact angle were made for the following liquids with different polarities: distilled water, diiodomethane, and ethylene glycol. During the determination of surface energy on each of the three samples of one material, 10 contact angles were made for each of the three liquids. The surface free energy was determined using software module SCA 20. 

#### 3.3.5. Thermal Analysis of PLA and PHA with Polyphenols

**Differential scanning calorimetry (DSC)** Using a Mettler Toledo DSC analyzer (Mettler-Toledo, Greifensee, Switzerland), the temperature ranges of the sample phase changes were determined, i.e., the glass transition temperature (T_g_), crystallization temperature (T_cc_), melting temperature of the crystalline phase (T_m_), and oxidation temperature (T_o_). The heat (ΔH) accompanying the phase changes was also determined. Test samples weighing 5 to 6 μg were placed in 100 μL open, aluminum crucibles. The samples were heated from 0 to 200 °C at a rate of 20 °C/min under an argon atmosphere. After 10 min at 200 °C, the samples were cooled to 0 °C. Then, the gas was switched from argon to air (flow rate 50 mL/min), and the samples were heated to 350 °C.

**Thermal decomposition (TGA)** This was determined by using a Mettler Toledo Thermobalance (Mettler-Toledo, Greifensee, Switzerland) to obtain the initial temperature of degradation and the maximum thermal degradation temperature. PLA and PHA samples with a weight of 5 mg were heated at a 10 °C/min from 25 to 600 °C under an inert N_2_ atmosphere.

**The Vicat softening temperature** This is the temperature at which a hardened steel needle with a circular cross-section of 1 mm^2^ penetrates the test sample to a depth of 1 under a standard load of 10 N. The Vicat mm softening temperature test was measured using a D-Vicat.HDT/3/400FA apparatus (Peter Huber Kältemaschinenbau GmbH, Offenburg, Germany). The measurement parameters were as follows: 10 N load, temperature gradient 120 °C/h, initial temperature 45 °C, final temperature 150 °C, and displacement range up to 15 mm. Displacement was measured with inductive displacement gauges with 0.001 mm resolution. The test pieces were square plates cut from extruded strip with a side length of 10 mm and a thickness of at least 1.6 mm. The sample consisted of a stack of 3 plates. The surfaces of the samples were parallel and smooth.

**The melt flow index test.** This was performed in accordance with ISO 1133D using a MeltFloWon Plus apparatus (CEAST, Planegg, Germany). The melt flow rate index (MFR, g/10min) and melt volume rate index (MVR, cm^3^/10 min) were determined. The measurement conditions were as follows: plasticizing temperature 190 °C, load of 2.160 kg, preheating time of 240 s, nozzle length of 8 mm, and nozzle diameter of 2.095 mm.

#### 3.3.6. Change of Color 

Change of color measurements were carried out using a CM-3600d spectrophotometer (Konica Minolta Sensing, Osaka, Japan). Color measurements were performed to determine the color change of the PLA and PHA samples after 100 h, 200 h and 300 h of UV exposure, thermal oxidation, and weathering. The result of the test is the color, as described in the CIE-Lab space and the color in a system of three coordinates, i.e., L, a, and b, where L is the lightness parameter (maximum value of 100, representing a perfectly reflecting diffuser, minimum value of zero representing the color black), a is the axis of red-green, and b is the axis of yellow-blue. The a and b axes have no specific numerical limits. The change of color, dE·ab, was calculated according to Equation (7):
(7)$$\mathrm{dE}\cdot \mathrm{ab}\;=\;\sqrt{\left(\Delta a^{2})+(\Delta b^{2})+(\Delta L^{2}\right)}$$

The visual changes of color of the PLA and PHA samples after aging were recorded using a camera.

For change of color determination, 3 samples of each type of the polymeric materials were prepared for each time interval of the respective types of aging. Spectrophotometric determinations were made at 5 measuring points of each of the 3 control samples. 

#### 3.3.7. Accelerated Aging of PLA and PHA with Polyphenols

**UV aging.** This was performed using a UV 2000 apparatus (Atlas Material Testing Technology LLC, USA). The measurement lasted 100 h, 200 h, and 300 h and consisted of two alternately repeating cycles with the following parameters: daily cycle (radiation intensity = 0.7 W/m^2^, temperature 60 °C, and duration 8 h) and night cycle (no UV radiation, temperature = 50 °C, and duration 4 h).

**Thermo-oxidation aging.** The samples were exposed to air at an elevated temperature (70 °C) for 100 h, 200 h, and 300 h in a dryer (Binder, Germany) with forced convection.

**Weathering aging.** This was carried out using a Weather-Ometer Ci 4000 (Atlas Material Testing Technology LLC, Chicago, IL, USA) with inner and outer filters of type S borosilicate glass. The test consisted of two variable cycles simulating daytime and night time conditions, and the samples were subjected to two different cycles as follows: day cycle (radiation intensity E = 40 W/m^2^ = 0.144 MJ/m^2^ over a λ range of 300 to 400 nm, temperature of 60 °C, duration of 240 min, humidity at 80%, rain water on), and night cycle (no radiation, temperature at 50 °C, humidity of 60%, duration of 120 min). Material weathering tests with a xenon-arc light were performed according to ISO 4892-2 (accelerated weathering simulates the damaging effects on materials and coatings of long-term outdoor exposure).

All aging tests were performed at intervals of 100 h, 200 h, and 300 h. After 200 h aging, especially weathering aging, the samples underwent significant degradation making it impossible to perform some tests, for example, determination of mechanical properties. After 300 h of aging, the samples crumbled by themselves, which made most tests impossible.

#### 3.3.8. Statistical analysis

All tests presented in the manuscript were performed on three control samples and the average results were shown in the manuscript. Calculations were made for the means and standard deviations of three independent samples (*n* = 3). Statistical analyses were performed for the comparison of the means, and then, performed using a Fischer LSD test (the significance level was set at *p* < 0.05).

## 4. Conclusions

The results presented suggest that natural polyphenols can be successfully used as functional additives for biodegradable polyesters. The plant substances analyzed showed a stabilizing effect on polymers, as evidenced by higher oxidation temperatures of samples with polydatin and (+)-catechin. The clear stabilizing effect and stopping degradation processes were also found based on the analysis of changes of surface free energy and analysis of the aging coefficient (K) of PLA and PHA with polydatin and (+)-catechin. The stabilizing effect of polyphenols is the result of their strong antioxidant properties and the ability to reduce free radicals that initiate the degradation reactions of polymeric materials. Moreover, plant polyphenols are substances of varying color. (+)-Catechin can be used simultaneously as a natural dye and aging indicator. In contrast, polydatin did not dye polyesters and did not change their organoleptic properties, and thus fulfilled the conditions for typical polymer stabilizers. However, polydatin, similar to (+)-catechin, can be successfully used as an indicator substance for biodegradable polyesters which, by changing color, can indicate the lifetime of the polymer material. The proposed materials based on environmentally friendly polymers and functional additives of plant origin can be interesting alternatives to currently used packaging materials, including intelligent packaging materials.

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
