# Peer review of "The Application of (+)-Catechin and Polydatin as Functional Additives for Biodegradable Polyesters"

_ijms, 2020, doi:10.3390/ijms21020414_

Round 1
Reviewer 1 Report
This manuscript presents the use of two natural polyphenols ((+)-catechin and polidatin) as functional additives for PLA and PHA. The stabilizing effect of polyphenols on PLA and PHA was evidenced. Moreover, ((+)-catechin can be used as a natural dye and aging indicator. In general, this is interesting to the readers. Publication with minor revision is recommended. Some questions/concerns are listed below:
The solubility of ((+)-catechin and polidatin) as functional additives in PLA and PHA matrix was characterized by SEM, as shown in Figure 2. It is however not clear to me how to conclude why a particular image shows soluble or insoluble situations. In Table 1, a significant decrease in tensile strength for PLA material and a significant increase for PHA material were observed after thermooxidation and UV aging for 100h. This was attributed to the change of crystalline. I would recommend to confirm this by doing DSC measurements for the samples after aging. In Figure 4, a decrease of MFI and MVR was found for PLA-based blends. Is this due to H-bonding effect? The DSC curves should be provided. It should also be explained how the oxidation temperature was defined and measured. Thermogravimetric analysis (TGA) can provide valuable information regarding thermal stability. I suggest perform the TGA measurements if the instrument is available. When the crystallinity was calculated, the melting enthalpy of 100% crystalline P(3-HB) was used. This was however a different polymer as the P(3,4-HB) in this paper, which basically makes the calculated values meaningless. Is the value for P(3,4-HB) available in the literature or from the supplier? If not, I think the calculation should be removed. It can be interesting for the readers if the authors can discuss their observations in correlation with the different molecular structures of the two polyphenols.Author Response
Institute of Polymer and Dye Technology
Technical University of Lodz
90-924 Lodz, ul Stefanowskiego 12/16, Poland
Tel.: +48 42 631 32 23, Fax: +48 42 636 25 43
December 9, 2019
International Journal of Molecular Sciences — Open Access Journal
Dear Professor,
We are resubmitting our revised paper entitled “The application of (+)-catechin and polidatin as functional additives for biodegradable polyesters” by Malgorzata Latos-Brozio and Anna Masek with a request to reconsider it for publication in " International Journal of Molecular Sciences”.
We have carefully considered the Editor and Reviewers' comments. The manuscript was revised exactly according to these comments. The list of responses to the reviewer’s comments and corrections made in the manuscript is attached.
The manuscript has not been previously published, is not currently submitted for review to any other journal, and will not be submitted elsewhere before a decision is made by this journal.
For correspondence please use the following information:
corresponding author: Anna Masek
Institute of Polymer and Dye Technology
Technical University of Lodz
90-924 Lodz, ul Stefanowskiego 12/16, Poland
Tel.: +48 42 631 32 93
Fax: +48 42 636 25 43
e-mail: anna.masek@p.lodz.pl
Yours sincerely,
PhD, Dsc Anna Masek
Answers to reviewer #1 comments
Reviewer #1: This manuscript presents the use of two natural polyphenols ((+)-catechin and polidatin) as functional additives for PLA and PHA. The stabilizing effect of polyphenols on PLA and PHA was evidenced. Moreover, ((+)-catechin can be used as a natural dye and aging indicator. In general, this is interesting to the readers. Publication with minor revision is recommended. Some questions/concerns are listed below:
The solubility of ((+)-catechin and polidatin) as functional additives in PLA and PHA matrix was characterized by SEM, as shown in Figure 2. It is however not clear to me how to conclude why a particular image shows soluble or insoluble situations.
Answer: We agree with the Reviewer's comment. SEM photos do not provide information on solubility or insolubility of (+)-catechins and polydatin in polymers but show topography of samples. The mistake has been corrected.
Reviewer #1: In Table 1, a significant decrease in tensile strength for PLA material and a significant increase for PHA material were observed after thermooxidation and UV aging for 100h. This was attributed to the change of crystalline. I would recommend to confirm this by doing DSC measurements for the samples after aging.
Answer: DSC measurements for the samples after thermooxidation aging has been added. Due to the short time to improve the manuscript, we were unable to perform DSC measurements for samples after all aging.
Reviewer #1: In Figure 4, a decrease of MFI and MVR was found for PLA-based blends. Is this due to H-bonding effect?
Answer: Yes, the decrease of MFI and MFR index of samples with polyphenols can be due to H-bonding effect. Intermolecular hydrogen bonds (from polyphenols) bring the molecules closer together, as a result of which the density of the material increases.
Reviewer #1:The DSC curves should be provided. It should also be explained how the oxidation temperature was defined and measured.
Answer: The DSC curves has been added. The oxidation temperature is the temperature at which an exothermic oxidation peak appears on the DSC curve. Table 2 gives the initial (onset) temperatures of the oxidation peaks.
Reviewer #1:Thermogravimetric analysis (TGA) can provide valuable information regarding thermal stability. I suggest perform the TGA measurements if the instrument is available.
Answer: We agree with the Reviewer. Thermogravimetric analysis (TGA) has been added.
Reviewer #1: When the crystallinity was calculated, the melting enthalpy of 100% crystalline P(3-HB) was used. This was however a different polymer as the P(3,4-HB) in this paper, which basically makes the calculated values meaningless. Is the value for P(3,4-HB) available in the literature or from the supplier? If not, I think the calculation should be removed. It can be interesting for the readers if the authors can discuss their observations in correlation with the different molecular structures of the two polyphenols.
Answer: According to the manufacturer Simag Holdings LTD (Hong Kong, China) the PHA polymer is P(3,4HB). The melting enthalpy of 100% crystalline P(3,4HB) is not available in literature and from supplier. We agree with the Reviewer that the value melting enthalpy of 100% crystalline PHB does not quite correspond to the value of melting enthalpy of 100% crystalline P(3,4HB) and the calculations contain a certain error resulting from these differences. However, the only value of melting enthalpy of 100% crystalline PHB 146 J/g is available in the literature as melting enthalpy for polymers from the PHAs group. The results of calculations of the degree of crystallinity were included in the publication for comparative purposes, however, the above considerations regarding value of melting enthalpy of 100% crystalline materials were emphasized.
In addition, observations in correlation with the different molecular structures of the two polyphenols have been added to the manuscript. Polydatin has a higher molecular weight (Mw 390.38 g/mol) then (+)-catechin (Mw 290.27 g/mol) and it may behave in the polymer matrix as nuclei.
Reviewer 2 Report
The presented manuscript reports on the polymeric materials with two additives: (+)-catechin and polidatin. Development of new materials that potentially can be used for packing is, nowadays, a novel trend in material science research. The study is very systematic, however, somewhat is lacking the statistical analysis, good explanation why and comparison with other already published results. My general remark is why these particular additives (despite the fact that nobody was using (+)-catechins and polidatin) in packing). Detailed remarks:
1) SEM images show only topography. From such image, we can only see how particular sample was prepared. We cannot see any information on the dispersion level of (+)-catechins and polidatin. How do you know that you don't have bi-phasic distribution (surface PLA and at the botton as a second layer (+)-catechins ? The change of the surface morphology can stem from the way of mixing or inappropriate sample storage ?
2) Mechanical properties are shown with the accuracy to 0.1 MPa ? How many samples were measured and what is the standard deviation ? Are obtained differences statistically valid ?
3) Aging coefficient - what is the accuracy level of its determination ? What is the meaning of K= 1.2 if 1 denotes sample resistant to deformation ?
4) Figure 3 - if K is a dimensionless parameter (this comes from the equation (2)) - what is the meaning of the arbitrary units in K ? Error bars are standard deviation ? standard error ? From how many samples ? This applies to all figures included in the manuscript. Statistical validation between samples should be included in the figures. If measurements follow normal distribution (normality test) then simple t-Student test can be applied. If distribution of the parameters is not normal one - e.g. Mann-Whitney test should be used.
5) Aging is carried out for 300h (means c.a. 12 days) - is this value of practical relevance ? What happens after this time ?
6) DSC data (Table 2) - the similar remarks as for mechanical data (Table 1).
7) Fig.6 - please include more detailed description of the presented spectra.
8) Fig. 7 - are the differences in free surface energy are statistically valid ? The authors mentioned that they used three different liquids but they do not included all data. Which model was used to calculate surface free energy ? Please be more specific and leave only the description which is relevant to presented data.
Author Response
Institute of Polymer and Dye Technology
Technical University of Lodz
90-924 Lodz, ul Stefanowskiego 12/16, Poland
Tel.: +48 42 631 32 23, Fax: +48 42 636 25 43
December 9, 2019
International Journal of Molecular Sciences — Open Access Journal
Dear Professor,
We are resubmitting our revised paper entitled “The application of (+)-catechin and polidatin as functional additives for biodegradable polyesters” by Malgorzata Latos-Brozio and Anna Masek with a request to reconsider it for publication in " International Journal of Molecular Sciences”.
We have carefully considered the Editor and Reviewers' comments. The manuscript was revised exactly according to these comments. The list of responses to the reviewer’s comments and corrections made in the manuscript is attached.
The manuscript has not been previously published, is not currently submitted for review to any other journal, and will not be submitted elsewhere before a decision is made by this journal.
For correspondence please use the following information:
corresponding author: Anna Masek
Institute of Polymer and Dye Technology
Technical University of Lodz
90-924 Lodz, ul Stefanowskiego 12/16, Poland
Tel.: +48 42 631 32 93
Fax: +48 42 636 25 43
e-mail: anna.masek@p.lodz.pl
Yours sincerely,
PhD, Dsc Anna Masek
Answers to reviewer #2 comments
Reviewer #2: The presented manuscript reports on the polymeric materials with two additives: (+)-catechin and polidatin. Development of new materials that potentially can be used for packing is, nowadays, a novel trend in material science research. The study is very systematic, however, somewhat is lacking the statistical analysis, good explanation why and comparison with other already published results. My general remark is why these particular additives (despite the fact that nobody was using (+)-catechins and polidatin) in packing). Detailed remarks:
1) SEM images show only topography. From such image, we can only see how particular sample was prepared. We cannot see any information on the dispersion level of (+)-catechins and polidatin. How do you know that you don't have bi-phasic distribution (surface PLA and at the botton as a second layer (+)-catechins ? The change of the surface morphology can stem from the way of mixing or inappropriate sample storage ?
Answer: We fully agree with the Reviewer's comment. SEM photos do not provide information on the dispersion level of (+)-catechins and polydatin but show topography of samples. The mistake has been corrected.
Reviewer #2: 2) Mechanical properties are shown with the accuracy to 0.1 MPa ? How many samples were measured and what is the standard deviation? Are obtained differences statistically valid ?
Answer: We agree with the Reviewer. The experimental part has been supplemented with information on the number of samples, statistics and experimental uncertainty.
Statistical analysis: All tests presented in the manuscript were performed on three control samples and the average results were shown in the manuscript. Calculations were made for the means and standard deviations of three independent samples (n=3). Statistical analysis was applied for the comparison of the means and then performed using a Fischer LSD test (the significance level was set at p<0.05).
Mechanical properties: to determine mechanical properties, six test samples were cut out from each of three control samples.
Reviewer #2: 3) Aging coefficient - what is the accuracy level of its determination ? What is the meaning of K= 1.2 if 1 denotes sample resistant to deformation ?
Answer: Thank you for the important comment. A value of K = 1.2 means that the sample is resistant to deformation.
The values of the aging factor K are in the range of 0-1 (-). The value above this range results from the measurement uncertainty of the device for determining mechanical properties (according to the manufacturer, the measurement error of the device can be up to 20%) and from statistical analysis.
Reviewer #2: 4) Figure 3 - if K is a dimensionless parameter (this comes from the equation (2)) - what is the meaning of the arbitrary units in K? Error bars are standard deviation? standard error? From how many samples? This applies to all figures included in the manuscript. Statistical validation between samples should be included in the figures. If measurements follow normal distribution (normality test) then simple t-Student test can be applied. If distribution of the parameters is not normal one - e.g. Mann-Whitney test should be used.
Answer: Aging coefficient K is a dimensionless parameter. The arbitrary units (a.u.) has been converted to dimensionless (-).
All tests presented in the manuscript were performed on three control samples and the average results were shown in the manuscript.
Depending on the determination, a different number of measurements were made on the three samples tested, e.g. five color measurements were made on each of the three samples from one material during the colour determination. During the determination of surface energy on each of the three samples of one material, 10 contact angles were made for each of the three liquids. For destructive tests, such as determination of mechanical properties and MFI index determination, three measurements were made (three samples from one material).
The error bars on all figures are standard deviation.
Reviewer #2: 5) Aging is carried out for 300h (means c.a. 12 days) - is this value of practical relevance ? What happens after this time ?
Answer: Aging was performed at intervals of 100h, 200h and 300h. After 200h aging, especially weathering aging, the samples underwent significant degradation making it impossible to perform some tests, e.g. determination of mechanical properties. After 300 hours of aging, the samples crumbled by themselves, which made most tests impossible.
Reviewer #2: 6) DSC data (Table 2) - the similar remarks as for mechanical data (Table 1).
Answer: According to the Reviewer’s suggestion under Table 2 (DSC data) indication of uncertainty has been added. In the case of thermal analysis (DSC Table 2 and TGA Table 3) the values specified by the apparatus manufacturer were given as the measurement uncertainty.
Reviewer #2: 7) Fig.6 - please include more detailed description of the presented spectra.
Answer: A more detailed description of the WAXD spectra is given.
Reviewer #2: 8) Fig. 7 - are the differences in free surface energy are statistically valid? The authors mentioned that they used three different liquids but they do not included all data. Which model was used to calculate surface free energy? Please be more specific and leave only the description which is relevant to presented data.
Answer: Surface free energy was determined by the method after Owens, Wendt, Rabel and Kaelble (OWRK). Details of the method and statistic analysis are given in the manuscript.
Reviewer 3 Report
I do not see this paper as it stand adding much to the scientific world, there is some food work and may be too many analyses but I do not think they the correct methods to test for stabilisation effect nor the dyes,
The idea of additives in the polymers is stabilise it, i.e. prevent the oxidation, I cannot see strong evidence of that happening here, the paper list many analyses which are irrelevant to testing the polymer stabilisation by the additives, hence they are invalid. If there is an evidence of the effect of the additive on the polymer, it would be they are facilitating the oxidation, as it is clearly demonstrated on Figure 8, where there is clear change in colour for the samples which has the additive while there was no change in the plank polymer.
I do not believe the results from this work support the claim of multifunctional scientific novelty “The application of polidatin and (+)-catechin as multifunctional additives for biodegradable polymers is a scientific novelty”, as any plant extracts would give dark colours to polymers if they were treated thermally, in addition to the stated above, there isn’t much of evidence of stabilisation effect of those additive on the tested polymer.
The other claim of “These natural polyphenols have not been used so far in polymer materials as stabilizers” I do not think this is correct too, here are some
DOI: 10.3390/polym11040669
DOI: 10.1016/j.polymdegradstab.2017.07.012
May be chemically identifying the products causing the change in colour could help the dye claim. There are other methods could be used to test the stabilisation effect of additives on polymers which could be performed on the instrumentation used here, one example is the oxidation induction time (OIT) on the DSC.
Author Response
Institute of Polymer and Dye Technology
Technical University of Lodz
90-924 Lodz, ul Stefanowskiego 12/16, Poland
Tel.: +48 42 631 32 23, Fax: +48 42 636 25 43
December 9, 2019
International Journal of Molecular Sciences — Open Access Journal
Dear Professor,
We are resubmitting our revised paper entitled “The application of (+)-catechin and polidatin as functional additives for biodegradable polyesters” by Malgorzata Latos-Brozio and Anna Masek with a request to reconsider it for publication in " International Journal of Molecular Sciences”.
We have carefully considered the Editor and Reviewers' comments. The manuscript was revised exactly according to these comments. The list of responses to the reviewer’s comments and corrections made in the manuscript is attached.
The manuscript has not been previously published, is not currently submitted for review to any other journal, and will not be submitted elsewhere before a decision is made by this journal.
For correspondence please use the following information:
corresponding author: Anna Masek
Institute of Polymer and Dye Technology
Technical University of Lodz
90-924 Lodz, ul Stefanowskiego 12/16, Poland
Tel.: +48 42 631 32 93
Fax: +48 42 636 25 43
e-mail: anna.masek@p.lodz.pl
Yours sincerely,
PhD, Dsc Anna Masek
Answers to reviewer #3 comments
Reviewer #3: I do not see this paper as it stand adding much to the scientific world, there is some food work and may be too many analyses but I do not think they the correct methods to test for stabilisation effect nor the dyes,
The idea of additives in the polymers is stabilise it, i.e. prevent the oxidation, I cannot see strong evidence of that happening here, the paper list many analyses which are irrelevant to testing the polymer stabilisation by the additives, hence they are invalid. If there is an evidence of the effect of the additive on the polymer, it would be they are facilitating the oxidation, as it is clearly demonstrated on Figure 8, where there is clear change in colour for the samples which has the additive while there was no change in the plank polymer.
I do not believe the results from this work support the claim of multifunctional scientific novelty “The application of polidatin and (+)-catechin as multifunctional additives for biodegradable polymers is a scientific novelty”, as any plant extracts would give dark colours to polymers if they were treated thermally, in addition to the stated above, there isn’t much of evidence of stabilisation effect of those additive on the tested polymer.
Answer: Thank you for this comment. The methodology of our research is similar to the methodology in other publications of concentrated polymer stabilization (Kirschweng, B.; Tatraaljai, D.; Foldes, E.; Pukanszky, B. Natural antioxidants as stabilizers for polymers. Polym. Degrad. Stabil. 2017, 145, 25-40).
The stabilization of polymers in this manuscript should be understood, among others as an increase in oxidation stability, as indicated by DSC results. The stabilization of polymers should also be understood as increasing the resistance of polymers to various types of aging, as shown in the determination of mechanical properties and determination of aging coefficients, changes in the surface energy of samples as well as changes in the parameter used in industrial practice - Vicat softening temperature.
We cannot agree with the Reviewer's statement "any plant extracts would give dark colors to polymers if they were treated thermally". Our experience from the study of compounds of plant origin indicates that some substances (curcumin, B-carotene) added to polymers behave just the opposite of what the Reviewer says - they discolour under the influence of temperature and other factors (Malgorzata Latos-Brozio and Anna Masek, Food and Chemical Toxicology, https://doi.org/10.1016/j.fct.2019.110975).
Moreover, there are many manuscripts in the scientific literature showing the possibilities of using plant-derived substances as natural antioxidants for various groups of polymers, e.g.:
Kirschweng, B.; Tatraaljai, D.; Foldes, E.; Pukanszky, B. Natural antioxidants as stabilizers for polymers. Polym. Degrad. Stabil. 2017, 145, 25-40.
Compounds of plant origin, in addition to significant antioxidant properties, are often coloured substances. These two properties of polyphenols can be used in polymer materials for their simultaneous stabilization and dyeing, e.g.:
Masek, A.; Latos, M.; Piotrowska, M.; Zaborski, M. The potential of quercetin as an effective natural antioxidant and indicator for packaging materials. Food Packag Shelf Life. 2018, 16, 51–58.
Masek, A. Flavonoids as Natural Stabilizers and Color Indicators of Ageing for Polymeric Materials. Polymers. 2015, 7, 1125-1144.
More and more scientific publications also show the potential of phytochemicals as indicator substances in smart packaging materials, e.g.:
Singh, S.; Gaikwad, K.K.; Lee, J.S. Anthocyanin – A Natural Dye for Smart Food Packaging Systems. Korean Journal of Packaging Science & Technology. 2018, 24(3), 167-180.
Tichoniuk, M.; Radomska, N.; Cierpiszewski, R. The Application of Natural Dyes in Food Freshness Indicators Designed for Intelligent Packaging. Studia Oeconomica Posnaniensia. 2017, 5(7), 19-34.
Zhang, J.; Zou, X.; Zhai, X.; Huang, X.W.; Jiang, C.; Holmes. M. Preparation of an intelligent pH film based on biodegradable polymers and roselle anthocyanins for monitoring pork freshness. Food Chem. 2019, 272, 306–312.
Latos, M.; Masek, A.; Zaborski, M. The potential of juglone as natural dye and indicator for biodegradable polyesters. P. I. Mech.Eng. L-J. Mat. 2019, 233(3), 276–285.
Medina-Jaramillo, C., Ochoa-Yepes, O., Bernal, C., Famá, L., 2017. Active and smart biodegradable packaging based on starch and natural extracts. Carbohydr. Polym. 176, 187–194. https://doi.org/10.1016/j.carbpol.2017.08.079.
Sanches-Silva, A., Costa, D., Albuquerque, T.G., Buonocore, G.G., Ramos, F., Castilho, M.C., Machado, A.V., Costa, H.S., 2014. Trends in the use of natural antioxidants in active food packaging: a review. Food Addit. Contam. A 31 (3), 374–395. https://doi. org/10.1080/19440049.2013.879215.
In the light of the cited scientific publications and trends in packaging materials, our research seems justified.
Reviewer #3:The other claim of “These natural polyphenols have not been used so far in polymer materials as stabilizers” I do not think this is correct too, here are some
DOI: 10.3390/polym11040669
DOI: 10.1016/j.polymdegradstab.2017.07.012
Answer: The authors meant that polydatin and (+)-catechin have not been used as polymer stabilizers, especially for biodegradable polyesters. The sentence has been completed and corrected. We apologize for the mistake. Of course, we agree with the Reviewer that natural polyphenols are used to stabilize polymers and there are many publications on this subject.
Reviewer #3:May be chemically identifying the products causing the change in colour could help the dye claim.
Answer: The colour change of substances of natural origin from the group of polyphenols (such as (+)-catechin and polydatin) occurs due to their oxidation. We agree with the reviewer. Chemically identifying the products causing the change in colour during the aging of polymers can provide new information on the mechanism of the stabilizing action of polyphenols in polymers. Thank you for the valuable comment. In the future, we will expand our research to analyse the chemical compounds that accompany polymer degradation and the colour change of natural substances.
Reviewer #3: There are other methods could be used to test the stabilisation effect of additives on polymers which could be performed on the instrumentation used here, one example is the oxidation induction time (OIT) on the DSC.
Answer: We fully agree with the Reviewer. The oxidation induction time (OIT) on the DSC is a very good method to assess the stabilizing effect of additives on polymers. In the manuscript, we decided to do DSC OIT experiments because we wanted to show all thermal transformations in the polymers tested, as well as determine the degree of crystallinity of materials. We did not make the traditional OIT measurement due to the difficult selection of isothermal oxidation conditions, i.e. we tested the temperature range of 200-400oC and at low temperatures the polymers did not oxidize, while in the higher ones the samples were burned immediately and it was not possible to distinguish the peaks from combustion and oxidation.
The DSC-OIT experiments were also made in other publications for PLA analysis (A. Jaszkiewicz, A. K. Bledzki, R. van der Meer, P. Franciszczak, A. Meljon, How does a chain-extended polylactide behave?: a comprehensive analysis of the material, structural and mechanical properties, Polym. Bull. (2014) 71:1675–1690).
Reviewer 4 Report
Plant polyphenol is widely existing in plants, which has potent antioxidant and free radical scavenging ability. In addition, plant polyphenols also have excellent ultraviolet (UV) absorption characteristics, especially in the high energy region. So plant polyphenols can be used as an anti-aging agent and sunscreen effective ingredients. Polyesters of polylactide (PLA) and polyhydroxyalkanoate (PHA) have good biodegradability and biocompatibility, and can be used as a biomedical material or biodegradable packaging material. It has become the most active research focus in the field of biological materials in recent years. In this manuscript, the authors use natural polyphenols of -(+)-catechin and polydatin as plant functional additives of stabilizers, dyes, and indicators of aging into biodegradable aliphatic polyesters, PLA and PHA, to obtain environmentally friendly materials. The design and operation of this manuscript are innovative and promising. The authors should address the following issues prior to consideration for publication.
There are many language spelling problems, e.g., “polidatin” (Line 2, Line 14, Line 15, Line 23, Line 61, Line 63, Line 64, Line 67, Line 68, Line 91, Line 98, Line 186, Line 195, Line 221, Line226……), should be “polydatin”; “thermooxidation” (Line 158, Line 172, Line 198, Line 201, Line 206, Table 1, Line 213, Line 222 ……), should be “Thermal oxidation”. Some of the graphic design quality need to be improved; the histogram color scheme, the format of the illustration, and labeling methods for statistical differences should be clearly and keep uniform in the whole manuscript. Background knowledge of the aging mechanism of polymeric materials of PLA and PHA should be supplemented. Plant polyphenols of -(+)-catechin and polydatin as plant functional stabilizers can improve the aging resistance properties under thermal oxidation or UV. The underline mechanism should be addressed. The abbreviations of some nouns should be used correctly, e.g., “Polyactide” (Line 32, Line48, Line 59, Line 190, Line 265, Line 283, Line 284, Line 297, Line 311, Line 312), should be “PLA”; “polyhydroxyalkanoate” (Line 34, Line 84, Line 189, Line 265, Line 286, Line 304, Line 310, Line 322), should be “PHA”; “differential scanning calorimetry” ( Line 270), should be “DSC”. The cited references format should keep uniform in the whole manuscript, e.g., the DOI information in some references should be deleted; page number information (Line 397, Line 404, Line 406, Line 457) should be completed.Author Response
Institute of Polymer and Dye Technology
Technical University of Lodz
90-924 Lodz, ul Stefanowskiego 12/16, Poland
Tel.: +48 42 631 32 23, Fax: +48 42 636 25 43
December 9, 2019
International Journal of Molecular Sciences — Open Access Journal
Dear Professor,
We are resubmitting our revised paper entitled “The application of (+)-catechin and polidatin as functional additives for biodegradable polyesters” by Malgorzata Latos-Brozio and Anna Masek with a request to reconsider it for publication in " International Journal of Molecular Sciences”.
We have carefully considered the Editor and Reviewers' comments. The manuscript was revised exactly according to these comments. The list of responses to the reviewer’s comments and corrections made in the manuscript is attached.
The manuscript has not been previously published, is not currently submitted for review to any other journal, and will not be submitted elsewhere before a decision is made by this journal.
For correspondence please use the following information:
corresponding author: Anna Masek
Institute of Polymer and Dye Technology
Technical University of Lodz
90-924 Lodz, ul Stefanowskiego 12/16, Poland
Tel.: +48 42 631 32 93
Fax: +48 42 636 25 43
e-mail: anna.masek@p.lodz.pl
Yours sincerely,
PhD, Dsc Anna Masek
Answers to reviewer #4 comments
Reviewer #4: Plant polyphenol is widely existing in plants, which has potent antioxidant and free radical scavenging ability. In addition, plant polyphenols also have excellent ultraviolet (UV) absorption characteristics, especially in the high energy region. So plant polyphenols can be used as an anti-aging agent and sunscreen effective ingredients. Polyesters of polylactide (PLA) and polyhydroxyalkanoate (PHA) have good biodegradability and biocompatibility, and can be used as a biomedical material or biodegradable packaging material. It has become the most active research focus in the field of biological materials in recent years. In this manuscript, the authors use natural polyphenols of -(+)-catechin and polydatin as plant functional additives of stabilizers, dyes, and indicators of aging into biodegradable aliphatic polyesters, PLA and PHA, to obtain environmentally friendly materials. The design and operation of this manuscript are innovative and promising. The authors should address the following issues prior to consideration for publication.
There are many language spelling problems, e.g., “polidatin” (Line 2, Line 14, Line 15, Line 23, Line 61, Line 63, Line 64, Line 67, Line 68, Line 91, Line 98, Line 186, Line 195, Line 221, Line226……), should be “polydatin”; “thermooxidation” (Line 158, Line 172, Line 198, Line 201, Line 206, Table 1, Line 213, Line 222 ……), should be “Thermal oxidation”.
Answer: “Polidatin” has been replaced by “polydatin”. “Thermooxidation” has been changed to “thermal oxidation”.
Reviewer #4:Some of the graphic design quality need to be improved; the histogram color scheme, the format of the illustration, and labeling methods for statistical differences should be clearly and keep uniform in the whole manuscript.
Answer: The quality of illustrations has been improved. Methods for determining statistical differences have been standardized throughout the manuscript.
Reviewer #4:Background knowledge of the aging mechanism of polymeric materials of PLA and PHA should be supplemented. Plant polyphenols of -(+)-catechin and polydatin as plant functional stabilizers can improve the aging resistance properties under thermal oxidation or UV. The underline mechanism should be addressed.
Answer: We agree with the Reviewer. Adding a mechanism will significantly increase the value of publications. The mechanism has been added.
Reviewer #4:The abbreviations of some nouns should be used correctly, e.g., “Polyactide” (Line 32, Line48, Line 59, Line 190, Line 265, Line 283, Line 284, Line 297, Line 311, Line 312), should be “PLA”; “polyhydroxyalkanoate” (Line 34, Line 84, Line 189, Line 265, Line 286, Line 304, Line 310, Line 322), should be “PHA”; “differential scanning calorimetry” ( Line 270), should be “DSC”.
Answer: The abbreviations of polylactide, polyhydroxyalkanoate and differential scanning calorimetry has been improved.
Reviewer #4:The cited references format should keep uniform in the whole manuscript, e.g., the DOI information in some references should be deleted; page number information (Line 397, Line 404, Line 406, Line 457) should be completed.
Answer: References have been unified. The cited articles from the MDPI publishing do not have page number information, the citation is in line with MDPI and ACS Style.
Round 2
Reviewer 2 Report
Major and important - please correct error analysis in Tables 1 and 2.
For example, description of the errors presented below Table 1 (standard deviations: TFmax ±4.0 MPa; EFmax ±0.3 %; TS ± 3.3 MPa; Eb ± 0.8%)
is not true ! This is very cardinal mistake.
1) Standard deviation should be calculated for each sample separately. If the quantity comes from measurement, to calculate the standard deviation:
SD = sum (all values)/n where n-number of measurements.
2) If final parameter value comes from the model or equation - you should used, for example, maximal error taking into account error propagation.
3) Writing e.g. 65.4 MPa has no sense if the error is 4.0 MPa. It is 65 +/- 4.0 MPa.
Minor:
Please check English once more before publishing, especially the use of was/were e.g. "All aging tests was performed" should be written as "All aging tests were performed"
Author Response
Institute of Polymer and Dye Technology
Technical University of Lodz
90-924 Lodz, ul Stefanowskiego 12/16, Poland
Tel.: +48 42 631 32 23, Fax: +48 42 636 25 43
December 18, 2019
International Journal of Molecular Sciences — Open Access Journal
Dear Professor,
We are resubmitting our revised paper entitled “The application of (+)-catechin and polidatin as functional additives for biodegradable polyesters” by Malgorzata Latos-Brozio and Anna Masek with a request to reconsider it for publication in " International Journal of Molecular Sciences”.
We have carefully considered the Editor and Reviewers' comments. The manuscript was revised exactly according to these comments. The list of responses to the reviewer’s comments and corrections made in the manuscript is attached.
The manuscript has not been previously published, is not currently submitted for review to any other journal, and will not be submitted elsewhere before a decision is made by this journal.
For correspondence please use the following information:
corresponding author: Anna Masek
Institute of Polymer and Dye Technology
Technical University of Lodz
90-924 Lodz, ul Stefanowskiego 12/16, Poland
Tel.: +48 42 631 32 93
Fax: +48 42 636 25 43
e-mail: anna.masek@p.lodz.pl
Yours sincerely,
PhD, Dsc Anna Masek
Answers to reviewer #2 comments (Round 2)
Reviewer #2: Major and important - please correct error analysis in Tables 1 and 2.
For example, description of the errors presented below Table 1 (standard deviations: TFmax ±4.0 MPa; EFmax ±0.3 %; TS ± 3.3 MPa; Eb ± 0.8%)
is not true ! This is very cardinal mistake.
1) Standard deviation should be calculated for each sample separately. If the quantity comes from measurement, to calculate the standard deviation:
SD = sum (all values)/n where n-number of measurements.
2) If final parameter value comes from the model or equation - you should used, for example, maximal error taking into account error propagation.
3) Writing e.g. 65.4 MPa has no sense if the error is 4.0 MPa. It is 65 +/- 4.0 MPa.
Answer: We fully agree with the Reviewer's comment. We apologize for our mistake. The error analysis in Tables 1 and 2 has been corrected as suggested by the Reviewer.
Reviewer 3 Report
As I understand it, the aim of the work was to use the natural polyphenols (polydatin and (+)-catechin) in the selected polymer materials as stabilizers. It is known that un-stabilised PHA has poor thermal stability (10.1038/am.2016.48). The presented results of the polymer with both polydatin and (+)-catechin are very similar to that of the un-stabilised polymers, hence there is no stabilisation effect.
Author Response
Institute of Polymer and Dye Technology
Technical University of Lodz
90-924 Lodz, ul Stefanowskiego 12/16, Poland
Tel.: +48 42 631 32 23, Fax: +48 42 636 25 43
December 18, 2019
International Journal of Molecular Sciences — Open Access Journal
Dear Professor,
We are resubmitting our revised paper entitled “The application of (+)-catechin and polidatin as functional additives for biodegradable polyesters” by Malgorzata Latos-Brozio and Anna Masek with a request to reconsider it for publication in " International Journal of Molecular Sciences”.
We have carefully considered the Editor and Reviewers' comments. The manuscript was revised exactly according to these comments. The list of responses to the reviewer’s comments and corrections made in the manuscript is attached.
The manuscript has not been previously published, is not currently submitted for review to any other journal, and will not be submitted elsewhere before a decision is made by this journal.
For correspondence please use the following information:
corresponding author: Anna Masek
Institute of Polymer and Dye Technology
Technical University of Lodz
90-924 Lodz, ul Stefanowskiego 12/16, Poland
Tel.: +48 42 631 32 93
Fax: +48 42 636 25 43
e-mail: anna.masek@p.lodz.pl
Yours sincerely,
Answers to reviewer #3 comments (round 2)
Reviewer #3: As I understand it, the aim of the work was to use the natural polyphenols (polydatin and (+)-catechin) in the selected polymer materials as stabilizers. It is known that un-stabilised PHA has poor thermal stability (10.1038/am.2016.48). The presented results of the polymer with both polydatin and (+)-catechin are very similar to that of the un-stabilised polymers, hence there is no stabilisation effect.
Answer: Thank you for your comment. Stabilization means not only increasing the temperature but also extending (temporal) oxidation time. Therefore, it cannot be clearly stated on the basis of temperature that there is no stabilizing effect.
In addition to the increased oxidation temperature of samples with polydatin and (+)-catechin (Table 2), also other tested parameters testify to the increased resistance of samples to degradation agents, e.g. the aging coefficient (K) of PLA and PHA with polydatin and (+)-catechin after thermal oxidation and UV aging (Figure 3). These results suggest that the addition of plant polyphenols increased the resistance of both polymers to thermo-oxidative and UV aging.
Reviewer 4 Report
Accept as is.
Author Response
Institute of Polymer and Dye Technology
Technical University of Lodz
90-924 Lodz, ul Stefanowskiego 12/16, Poland
Tel.: +48 42 631 32 23, Fax: +48 42 636 25 43
December 18, 2019
International Journal of Molecular Sciences — Open Access Journal
Dear Professor,
We are resubmitting our revised paper entitled “The application of (+)-catechin and polidatin as functional additives for biodegradable polyesters” by Malgorzata Latos-Brozio and Anna Masek with a request to reconsider it for publication in " International Journal of Molecular Sciences”.
We have carefully considered the Editor and Reviewers' comments. The manuscript was revised exactly according to these comments. The list of responses to the reviewer’s comments and corrections made in the manuscript is attached.
The manuscript has not been previously published, is not currently submitted for review to any other journal, and will not be submitted elsewhere before a decision is made by this journal.
For correspondence please use the following information:
corresponding author: Anna Masek
Institute of Polymer and Dye Technology
Technical University of Lodz
90-924 Lodz, ul Stefanowskiego 12/16, Poland
Tel.: +48 42 631 32 93
Fax: +48 42 636 25 43
e-mail: anna.masek@p.lodz.pl
Yours sincerely,
PhD, Dsc Anna Masek
Round 3
Reviewer 2 Report
I have no further question. All issues raised by me were improved in a satisfactory way.
Reviewer 3 Report
.